# Factors associated with breast cancer screening intention in Kathmandu Valley, Nepal

Divya Bhandari[1], Akira Shibanuma[1], Junko Kiriya[1], Suzita Hirachan[2], Ken Ing Cherng Ong[1]*, Masamine Jimba[1]

1 Department of Community and Global Health, Graduate School of Medicine, The University of Tokyo, Bunkyo, Japan, 2 Tribhuvan University Teaching Hospital, Kathmandu, Nepal

* kenicong@m.u-tokyo.ac.jp

## Abstract

### Background

Breast cancer burden is increasing in low-income countries (LICs). Increasing incidence and delayed presentation of breast cancer are mainly responsible for this burden. Many women do not participate in breast cancer screening despite its effectiveness. Moreover, studies are limited on the barriers associated with low utilization of breast cancer screening in LICs. This study identified breast cancer screening behavior and factors associated with breast cancer screening intention among women in Kathmandu Valley, Nepal.

### Methods

A cross-sectional study was conducted among 500 women living in five municipalities of Kathmandu Valley, Nepal. Data were collected from July to September 2018, using a structured questionnaire. Interviews were conducted among women selected through proportionate random household sampling. This study was conceptualized using the theory of planned behavior, fatalism, perceived susceptibility, and perceived severity. The outcome variables included: the intention to have mammography (MMG) biennially, the intention to have clinical breast examination (CBE) annually, and the intention to perform breast self-examination (BSE) monthly. Analysis was conducted separately for each outcome variable using partial proportional odds model.

### Results

Out of 500 women, 3.4% had undergone MMG biennially, 7.2% CBE annually, and 14.4% BSE monthly. Women with a positive attitude, high subjective norms, and high perceived behavioral control were more likely to have the intention to undergo all three screening methods. Similarly, women were more likely to have intention to undergo CBE and MMG when they perceived themselves susceptible to breast cancer. Conversely, women were less likely to have intention to undergo CBE when they had high fatalistic beliefs towards breast cancer.

**Data Availability Statement:** All data files are available from the Figshare Repository at https://doi.org/10.6084/m9.figshare.12442214.

**Funding:** This study is funded by The University of Tokyo Fund to MJ. The funder has no role in the study design, data collection and analysis, decision to publish, or preparation of the manuscript.

**Competing interests:** The authors have declared that no competing interests exist.

## Conclusion

Women in this study had poor screening behavior. The practice of breast self-examination was comparatively higher than clinical breast examination and mammography. Multidimensional culturally sensitive interventions are needed to enhance screening intentions. Efforts should be directed to improve attitude, family support, and fatalistic belief towards cancer. Furthermore, the proper availability of screening methods should be ensured while encouraging women to screen before the appearance of symptoms.

## Introduction

Breast cancer is a leading global health problem. There were 2.1 million cases diagnosed in 2018, and 627,000 died of breast cancer globally [1, 2]. Among women, it is the most commonly diagnosed cancer and the leading cause of cancer deaths. The mortality to incidence ratio of breast cancer in low-income countries (LICs) is three times higher than in high-income countries [2]. This highlights the increasing burden of breast cancer in LICs. Moreover, the incidence of breast cancer is increasing in LICs due to physical inactivity, changes in reproductive patterns, and unhealthy dietary habits [3, 4].

Breast cancer screening is an effective prevention strategy to reduce breast cancer burden [5]. Mammography (MMG), clinical breast examination (CBE), and breast self-examination (BSE) are three widely practiced screening tests. MMG is recommended as a standard screening test globally [5]. However, considering the cost-effectiveness, CBE, and BSE are also recommended for low-resource settings [6–8]. Early detection of breast cancer saves lives, preserves the quality of life, and prevents catastrophic out-of-pocket payments [9, 10]. In addition, early diagnosed cases can be successfully treated with less extensive breast conservative surgery [11].

However, the late presentation of breast cancer is still common, and it has worsened the economic and health conditions of LICs such as Nepal [12, 13]. Majority of women in LICs tend to seek medical treatment in the late stages of cancer [14]. According to a clinical study conducted in Nepal, most breast cancer patients sought treatment at late stages (stages II and III) with an average tumor size of two to five centimeters [15]. This delay has led to increased tumor size, complicated treatment, and finally, premature mortality in Nepal [16–18]. Further, the absence of universal health coverage adds a substantial financial burden to the patients and their families [19, 20].

Considering these detrimental consequences of breast cancer, it is necessary to identify factors associated with low screening intentions. So far, different reasons have been identified for low screening intentions, such as lack of education, absence of family history, poor access to screening, financial difficulties, and fear [21, 22]. However, a great part of variance remained unexplained due to the inclusion of limited factors in those studies [23]. This study builds on previous research by including relevant factors based on the concept of the theory of planned behavior (TPB), perceived susceptibility, perceived severity, and fatalism [23].

Furthermore, studies on breast cancer screening intention and behavior are limited in LICs. Therefore, this study aimed 1) to examine breast cancer screening behavior among women in Kathmandu Valley, Nepal, and 2) to identify the factors associated with the intentions to perform breast cancer screening tests (MMG, CBE, and BSE).

## Methods

### Settings and participants

A cross-sectional study was conducted among Nepalese women residing in Kathmandu Valley, Nepal. Proportional random sampling was used to recruit the participants in this study. Women were selected from five municipalities (Kathmandu, Bhaktapur, Lalitpur, Kirtipur, and Madhyapur Thimi) which are considered as a core urban area of the Valley [24]. The study was focused on these areas because the urban population is at higher risk of breast cancer, and screening services (especially mammography) are available in those areas compared to other areas of Nepal. From five municipalities, a total of 20 wards were selected randomly. A random household sampling was conducted proportionately to the number of households in those wards.

This study recruited women aged 40 years and above as eligible participants considering the age of women recommended to undergo mammography by the American Cancer Society [5]. Women with a history of breast cancer were excluded from the study.

The sample size was determined using the OpenEpi sample size calculator for a cross-sectional study with a power of 80%, and a confidence level of 95%. As reported in a previous study with a similar study design [25], our calculation was based on the assumption that 7.1% of people with good literacy and 1.8% of people with low literacy practiced breast cancer screening. The calculated minimum sample size was 476. Considering 10% dropout, the target sample size was increased to 529. Compare to other factors included in this study, literacy required a larger sample size to show association with the independent variable, so it was considered for calculating the sample size.

### Survey tools

Validated questionnaires from previous studies were used in this study [26–29]. Formal permissions were received from authors before using their questionnaire. The questionnaire unavailable in the Nepalese language was translated, back-translated, and further reviewed by experts. The expert panel included two breast cancer experts, two Nepali language experts, two public health experts, and three local leaders. It was then pretested among 50 women (around 10% of the target sample size) living in ward number 32 of Kathmandu district. The data collected from the pretest were not included in the main analysis. Inter-item reliability coefficients (Cronbach's alpha) were checked to ensure the internal consistency of items. The Cronbach's alpha for each construct included in the questionnaire was above 0.7, which is mentioned below. Some of the wordings were rephrased to make it understandable among local women.

### Exposure variables and assessment

This study included perceived susceptibility, perceived severity, and fatalism along with the constructs of the TPB (attitude, subjective norms, and perceived behavioral control) as exposure variables.

**Attitude, subjective norms, and perceived behavioral control (constructs of the TPB).** TPB is a health behavioral model, which presumes that attitudes towards behavior, subjective norms, and, perceived behavioral control, together determine the individual intention and behavior [30]. The questionnaire developed by Gaston Godin was used in this research to measure the attitudes towards behavior, subjective norms, and perceived behavioral control [26]. Each construct (attitude, subjective norms, and perceived behavioral control) consists of three items that were measured on a five-point Likert scale (1: strongly disagree/strongly oppose/

very difficult/very improbable to 5:strongly agree/ strongly favor/very easy/very probable). All three items were summed up to calculate the total scores (range 3 to15) for each construct. A higher score indicated positive/good attitudes, higher subjective norms, and higher perceived behavior, respectively. The Cronbach's alpha for this scale ranged from 0.79 to 0.91 in this study.

**Fatalism.** Fatalism is a psychological doctrine where individuals believe all events are fated, and human beings cannot change or control the outcomes [31, 32]. It was measured using the revised Powe fatalism inventory scale initially developed by Barbara Powe and later modified by Rachel Mayo [27]. The revised Fatalism scale consists of a total of 11 items (yes/no) grouped in four subscales: predetermination (five items), religiosity (one item), inevitable death (two items), and pessimism (three items). The total score for fatalism (range 0–11) was calculated by adding responses of all 11 items which were coded as yes = 1 and no = 0. Fatalism was treated as a continuous variable and its' higher value represented higher fatalism. The Cronbach's alpha for the fatalism scale was 0.88 in this study.

**Perceived susceptibility and severity.** Perceived susceptibility is the degree to which a person considers themselves susceptible to disease [33], and perceived severity is perception regarding the consequences of disease [34]. They were measured using the validated 'Nepalese Health Belief Model scale' [28]. It consists of five items to measure perceived susceptibility and seven items to measure perceived severity. Responses were rated on a five-point Likert scale (1 'strongly disagree' to 5 'strongly agree'). The total score for susceptibility was calculated by adding the scores of all five items. Likewise, the total score for severity was calculated by adding all seven items. A higher score meant higher susceptibility and higher severity. The Cronbach's alpha for susceptibility subscale was 0.97, and the severity subscale was 0.87 in this study.

## Outcome variable and assessment

**Breast cancer screening behavior.** The main outcome variable for the first objective of this study was breast cancer screening behavior. We asked participants whether they had undergone each screening method (MMG, CBE, and BSE) and assessed their screening behavior as a binary variable (yes/no). For those who answered yes, further follow-up question on frequency of screening was also asked.

**Intention to have breast cancer screening.** The main outcome variable for the second objective of this study was the intention to undergo each breast cancer screening methods. Participants reported their intentions to undergo MMG biennially, CBE annually, and BSE monthly on a five-point Likert scale (1 'strongly disagree' to 5 'strongly agree') each.

## Potential confounders and assessment

The following variables were also assessed as potential confounders.

**Knowledge of breast cancer.** Participants with higher knowledge of breast cancer are more likely to have a higher intention or probability of undergoing screening [34]. At the same time, knowledge can bring positive change in attitude and the fatalistic belief of the person [35]. Knowledge was measured using 21-item 'Modified Comprehensive Breast Cancer Knowledge Test' (yes/no/don't know) questionnaire, previously validated in the Nepalese context [29]. The correct response was scored '1' while incorrect answer or 'don't know' responses were scored '0'. The total score for knowledge was calculated by adding scores of all 21 items and it was treated as a continuous variable. The Cronbach's alpha for this scale was 0.83 in this study.

**Socio-demographic characteristics.** This study adopted socio-demographic variables from the Nepal Demographic Health Survey 2016 [36]. The variables included were age,

educational level, religion, ethnicity, current occupation, husband's education, husband's occupation, family income and time to the nearest screening facility. Age is considered as one of the factors to influence screening behavior and intention in the previous study. Also, with an increase in age, people are more likely to develop fatalistic beliefs towards cancer and might perceive themselves less susceptible to cancer [36, 37]. Other factors like education, income was associated with the screening behavior and intention in a previous study [38]. These factors are also more likely to influence perceived behavioral control and attitude towards screening [38]. Additionally, the family history of breast cancer, participation in an awareness program on breast cancer, and any family member from the health field was also asked (yes/no). Family history and family member from the health field is also likely to influence both attitude and behavior [39].

## Data collection

Data were collected through face-to-face interviews using a structured questionnaire from July to September 2018. Data collection was done by the principal researcher along with three trained research assistants with a public health background. Data collection in the Lalitpur municipality was also assisted by medical doctors from KIST medical college located in that municipality. The 30-minute long interview was conducted by visiting each household.

## Data analysis

Data were entered in Epi Data version 3.1 and exported to Stata version 13.1 for analyses. Data were analyzed using the partial proportional odds model (PPOM). Due to the ordinal nature of outcome variables, data were first fitted with the standard ordinal logistic regression (i.e. proportional odds model (POM)) model. However, POM was found inappropriate due to the violation of a proportional odds assumption for some independent variables. The proportional odds assumption was checked using a series of Wald tests, and Brant tests [40, 41]. The other alternatives were PPOM and fully unconstrained generalized ordered logit model (GOLM). PPOM relaxes the proportional odds assumptions for only those variables where it is violated. Whereas, GOLM relaxes the assumption for all variables, even if the assumption was violated by a few of them [42], resulting in too many parameters. Therefore, PPOM is usually considered a more efficient alternative to the GOLM [42]. Nonetheless, both PPOM and GOLM models were compared based on the Akaike Information Criterion (AIC) and Bayes' Information Criterion (BIC) [43], and the PPOM model was found to have smaller AIC and BIC statistics. We, therefore, used PPOM as the final model to assess the factors associated with the intention to undergo screening.

This study had three outcome variables (Intention to undergo MMG, CBE, and BSE) measured in 5 categories (1: 'Strongly Disagree' (SD), 2: 'Disagree'(D), 3: 'Neutral' (N), 4: 'Agree' (A), and 5: 'Strongly Agree' (SA). Therefore, a total of three separate PPOMs were fitted and analyzed. For the MMG model, all variables fulfilled the proportional assumption. However, in the case of the CBE model, the assumption was violated for the 'perceived behavioral control' variable; and for the BSE model, the assumption was violated for 'knowledge of breast cancer', 'attitude', 'subjective norms', and 'perceived behavioral control'. In PPOM, for variables that meet the proportional odds assumption, only one odd ratio is reported; and for variables that fail to meet the assumption, multiple odd ratios are reported. As a result, Table 5 presents multiple adjusted odd ratios(AOR) for the variables that violate the assumption which has been labeled and also noted in the footnote of the table using symbols 'a', 'b', and 'c' where, a = AOR for SD&D versus N, A and SA; b = AOR for SD&D or N versus A and SA; c = AOR for SD&D or N or A versus SA. Although the responses of outcome variables were

measured on 5-level categories, responses in the SD category were found to be limited i.e. less than 15 for all three outcomes (as evident from Table 4). In particular, cross-tabulation showed that there were no respondents for the SD category corresponding to some categories of included independent variables. Therefore, "SD" category was merged with "D" category (renamed as "SD&D" category) to reduce false precision and to improve the stability and generalizability of the results.

Furthermore, before running the multivariable analysis, the multicollinearity test between independent variables was checked using the Variance Inflation Factor (VIF). Variable having a VIF of 10 or higher were excluded from the analyses [44]. The maximum VIF was 3.5 among the included variables in the final models. Table 5 present the result of the final multivariable PPOM models with AOR. Adjusted variables include age, no of children, education, occupation, family income, family members from a health background, the time required to reach a nearby health facility, and family history of breast cancer. Statistical significance for the final model was set at p<0.05.

### Ethics

This study obtained approval from the Research Ethics Committee of the Graduate School of Medicine, the University of Tokyo, Japan (SN 12034), and also from the Ethical Review Board of Nepal Health Research Council (SN 339/2018). The site approval letter was obtained from concerned district public health offices and metropolitan offices. Before data collection, informed written consent was obtained from the women participating in the study. Women participated voluntarily and their identity was kept anonymous by using identification codes.

## Results

Out of 529 women who were approached for this study, 500 agreed to participate. There was no missing data and all data from the 500 participants were analyzed in this study. Table 1 summarizes the socio-demographic characteristics of the women included in this study. The mean age of women was 48 years (standard deviation [SD] 5.5, range 40–69). Of the total, 18.0% had an education level of bachelor's degree and above. The median monthly family income was Nepalese Rupee (NPR) 47,500 ranging from NPR 825 to NPR 298,397. Of the total, 44.8% were housewives. Around 20% of women had a family history of breast cancer. Of the total, 37.4% had family members from the health field (students or professionals). Only 15% of the women participated in the awareness program related to breast cancer.

Table 2 shows a summary of the exposure variables. Among all variables, the mean score of perceived susceptibility towards breast cancer was the lowest with a score of 2.5 (SD 1.0).

Table 3 illustrates the practice of breast cancer screening. Of the total, 3.4% of women had undergone MMG biennially, 7.2% had undergone CBE annually, and 14.4% practiced BSE monthly. Similarly, Table 4 shows the breast cancer screening intention. Around 20% of women expressed strong intention (strongly agree) to undergo BSE. More than half (64.6%) of women disagreed of having the intention to undergo MMG.

Table 5 shows the result of the multivariable partial proportional odds model of the factors associated with breast cancer screening intention. After adjusting for confounders and other variables, women who participated in the awareness program of breast cancer were more likely to have the intention to undergo MMG (AOR = 2.69, 95% CI 1.42–5.11).

Women who perceived themselves susceptible to breast cancer were more likely to have the intention to undergo MMG (AOR = 1.06, 95% CI 1.01–1.12) and CBE (AOR = 1.08, 95% CI 1.03–1.13). In contrast, women with high fatalistic beliefs were less likely to have the intention to undergo CBE (AOR = 0.92, 95% CI 0.86–0.99).

**Table 1. Socio-demographic characteristics of participants (n = 500).**

| Socio-demographic variables | n | % |
|---|---|---|
| **Age\* [mean (SD)]** | 48.2 (5.5) | |
| **Ethnicity** | | |
| Brahmin/Chhetri | 264 | 52.9 |
| Janajati | 220 | 44.0 |
| Others (Dalit/Muslim/Madheshi) | 16 | 3.2 |
| **Religion** | | |
| Hindu | 466 | 93.2 |
| Buddhism/Islam/Christianity | 34 | 6.8 |
| **Education** | | |
| Illiterate | 52 | 10.4 |
| Can read and write only | 83 | 16.6 |
| Primary | 58 | 11.6 |
| Secondary | 136 | 27.2 |
| Higher secondary | 81 | 16.2 |
| Bachelor's degree and above | 90 | 18.0 |
| **Occupation** | | |
| Housewife | 224 | 44.8 |
| Business | 208 | 41.6 |
| Labor work | 16 | 3.2 |
| Services (Govt/private) | 52 | 10.4 |
| **Husband education (n = 475)** | | |
| Illiterate | 21 | 4.4 |
| Can read and write only | 22 | 4.6 |
| Primary | 43 | 9.1 |
| Secondary | 116 | 24.4 |
| Higher secondary | 88 | 61.1 |
| Bachelor's degree and above | 185 | 38.9 |
| **Husband's occupation (n = 475)** | | |
| Agriculture | 24 | 5.1 |
| Business | 156 | 32.8 |
| Labor work | 42 | 8.8 |
| Services (Govt/private) | 168 | 35.4 |
| Retried / foreign employment | 85 | 17.9 |
| **Monthly family Income (Median)** (1US$ = 110 NPR) | | NPR 47,500 |
| **Time taken to reach the nearest health facility** | | |
| Less than 30 min | 419 | 83.8 |
| 30 minutes and more | 81 | 16.2 |
| **Family history of breast cancer** | | |
| Yes | 104 | 20.8 |
| No | 396 | 79.2 |
| **Participation in any breast cancer training/awareness program** | | |
| Yes | 73 | 14.6 |
| No | 427 | 85.4 |
| **Family member from health field (student, health worker)** | | |
| Yes | 187 | 37.4 |
| No | 313 | 62.6 |

**Table 2. Summary table of exposure variables.**

| Exposure variables | Mean | SD |
|---|---|---|
| **Attitude** | | |
| Mammography | 3.7 | 0.5 |
| Clinical breast examination | 3.5 | 0.6 |
| Breast self-examination | 3.8 | 0.8 |
| **Subjective norm** | | |
| Mammography | 3.4 | 0.7 |
| Clinical breast examination | 3.3 | 0.8 |
| Breast self-examination | 3.4 | 1.2 |
| **Perceived behavioral control** | | |
| Mammography | 3.5 | 0.8 |
| Clinical breast examination | 3.5 | 0.6 |
| Breast self-examination | 2.8 | 1.4 |
| **Risk perception** | | |
| Perceived susceptibility | 2.5 | 1.0 |
| Perceived severity | 3.6 | 1.0 |
| Knowledge of breast cancer (Range: 0–21) | 9.9 | 3.5 |
| Breast cancer fatalism (Range: 0–11) | 4.8 | 3.5 |

For MMG, women with a positive attitude towards MMG (AOR = 1.40, 95% CI 1.19–1.65), higher subjective norms (AOR = 2.18, 95% CI 1.81–2.62), and high perceived behavioral control (AOR = 1.96, 95% CI 1.65–2.34) were more likely to have the intention to undergo MMG.

Similarly, women who had positive attitudes towards CBE (AOR = 1.25, 95% CI 1.11–1.41) and had higher subjective norms (AOR = 1.64, 95% CI 1.43–1.89) were more likely to have the intention to undergo CBE. Having a higher perceived behavior control was associated with the likelihood of being in a higher agreement level (A and SA) to undergo CBE as opposed to being at neutral or below neutral level (AOR = 1.55, 95% CI 1.24–1.95). The effects became much stronger with increment in perceived behavior control, further, the largest effect was identified among the final level (i.e. SA versus A, N or SD&D).

**Table 3. Breast cancer screening behavior (N = 500).**

| Behavior of breast cancer screening | n | % |
|---|---|---|
| **MMG** | | |
| Never | 448 | 89.6 |
| Occasionally | 35 | 7.0 |
| Biennially | 17 | 3.4 |
| **CBE** | | |
| Never | 400 | 80.0 |
| Occasionally | 64 | 12.8 |
| Annually | 36 | 7.2 |
| **BSE** | | |
| Never | 293 | 58.6 |
| Occasionally | 135 | 27.0 |
| Monthly | 72 | 14.4 |

MMG: mammography, CBE: clinical breast examination, BSE: breast self-examination.

**Table 4. Breast cancer screening intention (N = 500).**

| Intention to do breast cancer screening | Strongly Agree n (%) | Agree (%) | Neutral n (%) | Disagree n (%) | Strongly Disagree n (%) |
|---|---|---|---|---|---|
| MMG (biennially) | 32 (6.4) | 121 (24.2) | 24 (4.8) | 323 (64.6) | - |
| CBE (annually) | 36 (7.2) | 139 (27.8) | 32 (6.4) | 284 (56.8) | 9 (1.8) |
| BSE (monthly) | 99 (19.8) | 235 (47.0) | 80 (16.0) | 72 (14.4) | 14 (2.8) |

MMG: mammography, CBE: clinical breast examination, BSE: breast self-examination.

In the case of BSE, women with higher attitude (AOR = 2.91, 95% CI 2.13–3.99), and those with better subjective norms (AOR = 1.68, 95% CI 1.19–2.37) were positively associated with increased odds of expressing higher agreement (A and SA) to undergo BSE rather than expressing neutral or disagreement. Women who had high perceived behavioral control were more likely to have intention to undergo BSE (AOR = 2.11, 95% CI 1.69–2.62). Findings revealed that having better knowledge of breast cancer increased the odds of a woman expressing higher agreement to undergo BSE rather than expressing neutral or disagreement (A and SA versus N, D&SD; AOR = 1.18, 95% CI 1.08–1.29), though it failed to achieve statistical significance in other categories.

**Table 5. Factors associated with breast cancer screening intention (N = 500).**

| Variables | MMG | | CBE | | BSE | |
|---|---|---|---|---|---|---|
| | AOR | (95% CI) | AOR | (95% CI) | AOR | (95% CI) |
| Participation in awareness programs | | | | | | |
| No | 1.00 | | 1.00 | | 1.00 | |
| Yes | 2.69 ** | (1.42–5.11) | 1.72 | (0.92–3.19) | 1.69 | (0.75–3.85) |
| Fatalism | 0.96 | (0.88–1.05) | 0.92 * | (0.86–0.99) | 1.00 | (0.93–1.08) |
| Susceptibility | 1.06 * | (1.01–1.12) | 1.08 ** | (1.03–1.13) | 1.04 | (0.99–1.10) |
| Knowledge of breast cancer | 1.01 | (0.92–1.12) | 1.04 | (0.96–1.14) | 0.96 [a] | (0.88–1.05) |
| | | | | | 1.18 [b] *** | (1.08–1.29) |
| | | | | | 0.86 [c] | (0.72–1.02) |
| Attitude towards behaviour | 1.40 *** | (1.19–1.65) | 1.25 *** | (1.11–1.41) | 2.23[a] *** | (1.67–2.97) |
| | | | | | 2.91[b] *** | (2.13–3.99) |
| | | | | | 5.51[c] *** | (2.04–14.86) |
| Subjective norms | 2.18 *** | (1.81–2.62) | 1.64 *** | (1.43–1.89) | 1.44 [a] | (0.98–2.14) |
| | | | | | 1.68 [b] ** | (1.19–2.37) |
| | | | | | 13.13 [c] *** | (5.79–29.79) |
| Perceived behaviour control | 1.96 *** | (1.65–2.34) | 1.47 [a] *** | (1.19–1.81) | 2.11 *** | (1.69–2.62) |
| | | | 1.55 [b] *** | (1.24–1.95) | | |
| | | | 4.66 [c] *** | (3.10–7.01) | | |

*p<0.05,

**p<0.01,

***p<0.001 (Adjusted for age, no of children, education, occupation, family income, family members from a health background, time required to reach a nearby health facility, and family history of breast cancer) AOR: adjusted odds ratio, CI: confidence interval, MMG: mammography, CBE: clinical breast examination, BSE: breast self-examination. Dependent variable coding: SD&D = strongly disagree and disagree, N = neutral, A = agree, SA = strongly agree.

[Note: Only one set of AOR is presented for explanatory variables that meet the proportional odds assumption. For variables with non-proportional odds, three AORs are presented as symbolized by 'a', 'b', and 'c' where, a = AOR for SD&D versus N, A and SA; b = AOR for SD&D or N versus A and SA; c = AOR for SD&D or N or A versus SA.]

## Discussion

All three components of TPB (Attitude, subjective norms, and perceived behavior control) were positively associated with the intention to undergo MMG, CBE, and BSE. Similarly, women who perceived themselves susceptible to breast cancer were more likely to have the intention to undergo MMG and CBE. Women who participated in the breast cancer awareness program were more likely to have the intention to undergo MMG. In contrast, women with fatalistic beliefs were less likely to have the intention to undergo CBE. Furthermore, with an increase in knowledge of breast cancer, women were more likely to have the intention to undergo BSE.

Our study findings align with the previous studies that have used TPB to predict different screening and healthy behavior intentions [45–47]. In our study, women were more likely to have the intention to undergo screening tests when they had a positive attitude towards a particular test. Positive attitude was found as an important factor to change behavior like adopting a healthy lifestyle [48]. People intend and are motivated to do such behavior which they believe can lead to positive outcomes [49]. Furthermore, in our study, women receiving screening suggestions from their family members and close ones were more likely to have intention to do it. People usually adhere to the advice given by trusted family members and close ones [50]. Social ties were found to have a positive influence in bringing healthy changes among participants in different settings [51–53]. Therefore, educating a person alone is not sufficient; the involvement of friends and family (social network) is also salient to effectively promote screening intention. It is particularly important for countries like Nepal, where women need permission from their husbands and in-laws before making any personal decision or health choices [54–56].

Likewise, screening intention was higher among women who believed in their capability to go for screening and perceived fewer barriers to screening. Similar to our finding, in a study conducted among Latina women in the United States, women expressed greater intention of receiving cervical cancer screening tests when they had high perceived behavior control [57].

It is worth considering particularly in low and middle-income countries like Nepal where screening services are not easily available. Many women are not financially competent to afford expensive services like MMG. Women should be involved in income-generating activities where they could support their expenses. Besides, a screening test like BSE can be done by a woman herself under complete privacy and autonomy. Proper training and awareness must be provided so that women perceived control to undergo those screenings which are free of cost and can be done independently. Evidently, in our study women who had high knowledge of breast cancer were more likely to have the intention to undergo BSE.

Another factor associated with the intention to undergo screening was perceived susceptibility. Women were more likely to have the intention to undergo MMG and CBE when they perceived themselves susceptibility to breast cancer. This is also an important factor to be considered particularly for low and middle-income countries like Nepal where people seek services only when they perceive themselves susceptible to health risks [58, 59]. Women did not perceive the need of undergoing screening until they had recognizable symptoms of breast cancer [60]. As a result, victims end up facing multiple challenges associated with late presentation, treatment difficulties, financial cost, etc. It is pivotal to make people realize that anyone can suffer from breast cancer and their timely screening practice can prevent them from other detrimental consequences. Conducting an awareness program is therefore key to promote screening behavior such as MMG which needs a more conscious decision. Evidently in our study, women who had participated in the awareness program were more likely to have the intention to undergo MMG. In our study intention to undergo BSE was not associated with

perceived susceptibility. This could be because people usually prefer to seek help from health professionals when they actually perceived some kind of health threat.

Like perceived susceptibility, fatalism is a potential barrier preventing people from participation in health-promoting behaviors [61]. However, in our study, the fatalistic belief was associated with the intention to undergo CBE only. Women with high fatalistic beliefs were less likely to have the intention to undergo CBE. Previous studies have also presented the mixed result of fatalism in screening intentions [62, 63]. Therefore, there is a need for an in-depth exploration of this belief before making any concrete conclusion. Nonetheless one of the reasons for our findings could be because women in a religious country like Nepal are usually reluctant to show their private body parts to anyone including health professionals unless it is very essential. Unlike BSE (which is self-examination) and MMG (test conducted using a device), CBE is the manual palpation done by the health care professionals. Considering the nature of the test, women with religious and fatalistic beliefs might be reluctant to pursue it [64, 65]. Therefore, deep traditional and fatalistic beliefs should not be ignored while designing screening interventions.

This study has several limitations. First, women might have over-reported their intention to undergo and behavior of screening to avoid awkwardness due to further questioning on their intention and behavior. However, the chance of over-reporting is equal for all participants, so the association between the exposure variables and the outcome variables should not be affected by this bias. The study was conducted in the urban areas of Nepal. Therefore, it cannot be generalized to the entire population. However, urban areas were selected considering the availability and accessibility of the screening tests. Finally, women who did not agree to participate in this study might have a different attitude, fatalistic belief, and knowledge level, which is not reflected in this study.

Despite these limitations, this study provides a comprehensive understanding of different factors associated with breast cancer screening intention. This could be helpful to develop a culturally sensitive intervention to promote breast cancer screening in resource-limited settings. This study also provides information on breast cancer screening behaviors, knowledge level, and fatalistic beliefs of women towards breast cancer. Findings can pave a way for future studies on breast cancer screening behavior.

## Conclusion

This study revealed poor screening behavior of women living in Kathmandu Valley, Nepal. Findings highlight the importance of TPB (positive attitude, subjective norms, perceived behavior control) along with perceived susceptibility and fatalism in increasing intention to undergo screening tests. Thus, educating only an individual is not enough, the inclusion of family members and addressing deep fatalistic beliefs are crucial for the successful promotion of screening. Women should be encouraged to undergo screening timely even before the appearance of symptoms. Meanwhile, screening tests should be made available and approachable before advocating for those services. Most importantly, practical training on BSE should be provided so that women feel competent to carry out themselves. To conclude, multidimensional culturally sensitive interventions are necessary to promote breast cancer screening in Nepal.

## Supporting information

**S1 Checklist. STROBE checklist.**
(DOC)

## Acknowledgments

We would like to express our gratitude to all the women who participated in this study. We are grateful to every health professional, professors, local leaders, and experts for their contribution to questionnaire review and support during data collection. We would also like to thank Nepalese organizations and communities for their support and encouragement.

## Author Contributions

**Conceptualization:** Divya Bhandari, Masamine Jimba.

**Data curation:** Divya Bhandari, Akira Shibanuma, Ken Ing Cherng Ong, Masamine Jimba.

**Formal analysis:** Divya Bhandari, Akira Shibanuma, Ken Ing Cherng Ong.

**Funding acquisition:** Divya Bhandari, Masamine Jimba.

**Investigation:** Divya Bhandari, Suzita Hirachan, Ken Ing Cherng Ong, Masamine Jimba.

**Methodology:** Divya Bhandari, Junko Kiriya, Ken Ing Cherng Ong, Masamine Jimba.

**Project administration:** Divya Bhandari, Ken Ing Cherng Ong, Masamine Jimba.

**Resources:** Divya Bhandari, Akira Shibanuma, Junko Kiriya, Suzita Hirachan, Ken Ing Cherng Ong, Masamine Jimba.

**Software:** Divya Bhandari, Akira Shibanuma, Ken Ing Cherng Ong, Masamine Jimba.

**Supervision:** Akira Shibanuma, Junko Kiriya, Ken Ing Cherng Ong, Masamine Jimba.

**Validation:** Divya Bhandari, Suzita Hirachan, Ken Ing Cherng Ong, Masamine Jimba.

**Visualization:** Divya Bhandari, Ken Ing Cherng Ong.

**Writing – original draft:** Divya Bhandari.

**Writing – review & editing:** Divya Bhandari, Akira Shibanuma, Junko Kiriya, Ken Ing Cherng Ong, Masamine Jimba.

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
