## [Decision Letter · Decision Letter 0]

14 Jul 2020

PONE-D-20-17423

Factors associated with breast cancer screening intention in Kathmandu Valley, Nepal

PLOS ONE

Dear Dr. Ong,

Thank you for submitting your manuscript to PLOS ONE. After careful consideration, we feel that it has merit but does not fully meet PLOS ONE’s publication criteria as it currently stands. Therefore, we invite you to submit a revised version of the manuscript that addresses the points raised during the review process.

We look forward to receiving your revised manuscript.

Kind regards,

Amir H. Pakpour, Ph.D.

Academic Editor

PLOS ONE

Journal Requirements:

2. Please address the following:

a) Please include additional information regarding the survey or questionnaire used in the study and ensure that you have provided sufficient details that others could replicate the analyses. For instance, if you developed a questionnaire as part of this study and it is not under a copyright more restrictive than CC-BY, please include a copy, in both the original language and English, as Supporting Information. In addition, please provide details as to where the women were recruited from who participated in the pre-testing of this tool.

b) Please state whether or not the "expert panel" provided written, informed consent for their inclusion in this study concerning questionnaire development.

c) Please refer to any sample size calculations performed prior to participant recruitment. If these were not performed please justify the reasons. Please refer to our statistical reporting guidelines for assistance (https://journals.plos.org/plosone/s/submission-guidelines.#loc-statistical-reporting).

Reviewers' comments:

Reviewer's Responses to Questions

**Comments to the Author**

1. Is the manuscript technically sound, and do the data support the conclusions?

Reviewer #1: Yes

2. Has the statistical analysis been performed appropriately and rigorously? 

Reviewer #1: No

3. Have the authors made all data underlying the findings in their manuscript fully available?

Reviewer #1: Yes

4. Is the manuscript presented in an intelligible fashion and written in standard English?

Reviewer #1: Yes

5. Review Comments to the Author

Reviewer #1: Dear Author(s),

This manuscript is overall reasonable. Your look on this issue is appreciable and your effort is admirable, short of a few points recommended in below:

1. The number of keywords is high. Please apply MeSH part on pubmed.com and extract the most related keywords.

2. Please specify and account for type of sampling obviously. As well, the grounds of why choosing this kind of sampling. Why are not cluster sampling opted?

3. In method part, 3 outcomes were numbered and after that it was mentioned using multiple and logistic regression models. If there are 3 outcomes (responses), multivariate models should be applied, otherwise, 3 outcomes should enter in each model one at time or 3 items together comprise one item (outcome or response). I know what you mean, however, I anticipate you to edit this part of abstract.

4. In Method part-Survey tools, please report Cronbach’s alpha (structured questionnaire).

5. In Outcome variable and assessment- Intention to have breast cancer screening part, why was this variable transformed to binary outcome? You will lose the main information with this action. Please do not change this variable and use ordinal logistic regression model unless there is a valid study which previously indicated you can transform this outcome likewise Potential confounders and assessment- Knowledge of breast cancer. Please do not transform the ordinal variables to binary variables. This could reduce the validity of the results of analysis.

6. I did not understand what you mean? ” Additional support was also provided by political leaders and community workers of the respective areas” please omit this part.

7. Please transmit Ethics and Data analysis part before introduction of your variables and avoid many subtitles for Method parts. This can cause misleading.

8. Is there any test to display outlier or influential data? Please report. Moreover please check the pre-assumptions of logistic models.

9. The categories of patients’ educational level could be misleading. Higher secondary and above is not a reasonable cut off for this category. Either this variable should be retained continuous (with the years of education) or divided into more categories to find clear results.

10. Please state why median criterion reported for Monthly Income covariate.

11. Please interpret the results using the odds ratio and the coefficients affected significantly the responses.

12. There is not any path diagram or (DAG) and figure to illustrate the results. Please confirm the results with figures and diagram.

Regards,

Maryam Ganji

6. PLOS authors have the option to publish the peer review history of their article (what does this mean?). If published, this will include your full peer review and any attached files.

Reviewer #1: **Yes: **Maryam Ganji

---

## [Author Response · Author response to Decision Letter 0]

3 Aug 2020

Response attached as a file named Response to Reviewers.docx

---

## [Decision Letter · Decision Letter 1]

18 Aug 2020

PONE-D-20-17423R1

Factors associated with breast cancer screening intention in Kathmandu Valley, Nepal

PLOS ONE

Dear Dr. Ong,

Thank you for submitting your manuscript to PLOS ONE. After careful consideration, we feel that it has merit but does not fully meet PLOS ONE’s publication criteria as it currently stands. Therefore, we invite you to submit a revised version of the manuscript that addresses the points raised during the review process.

We look forward to receiving your revised manuscript.

Kind regards,

Amir H. Pakpour, Ph.D.

Academic Editor

PLOS ONE

Reviewers' comments:

Reviewer's Responses to Questions

**Comments to the Author**

1. If the authors have adequately addressed your comments raised in a previous round of review and you feel that this manuscript is now acceptable for publication, you may indicate that here to bypass the “Comments to the Author” section, enter your conflict of interest statement in the “Confidential to Editor” section, and submit your "Accept" recommendation.

Reviewer #1: (No Response)

2. Is the manuscript technically sound, and do the data support the conclusions?

Reviewer #1: Yes

3. Has the statistical analysis been performed appropriately and rigorously? 

Reviewer #1: No

4. Have the authors made all data underlying the findings in their manuscript fully available?

Reviewer #1: Yes

5. Is the manuscript presented in an intelligible fashion and written in standard English?

Reviewer #1: Yes

6. Review Comments to the Author

Reviewer #1: Dear author(s),

Many thanks to edit your manuscript slightly, however, I highly anticipate you to redo your statistical analysis. It is vital that your result reflect whole study. Unfortunately, I did not convince about your results, so I return your manuscript again and write new comment, though they are repetitive.

1. Why have keywords been eliminated?

2. In Outcome variable and assessment- Intention to have breast cancer screening part, why was this variable transformed to binary outcome? You will lose the main information with this action. Please do not change this variable and use ordinal logistic regression model unless there is a valid study which previously indicated you can transform this outcome likewise Potential confounders and assessment- Knowledge of breast cancer. Please do not transform the ordinal variables to binary variables. This could reduce the validity of the results of analysis and please check the pre-assumptions of logistic models.

3. Please transmit Ethics and Data analysis part before introduction of your variables and avoid many subtitles for Method parts. This can cause misleading.

4. Please state why median criterion reported for Monthly Income covariate.

5. Please interpret the results using the odds ratio and the coefficients affected significantly the responses.

6. There is not any path diagram or (DAG) and figure to illustrate the results. Please confirm the results with figures and diagram.

Regards,

Maryam Ganji

7. PLOS authors have the option to publish the peer review history of their article (what does this mean?). If published, this will include your full peer review and any attached files.

Reviewer #1: No

---

## [Author Response · Author response to Decision Letter 1]

14 Sep 2020

Dear Reviewer,

We highly appreciate your meticulous review and constructive comments and suggestion. Probably due to some technical issues, you were not able to receive our initial response to the first round of comments. We are extremely sorry for the inconvenience. Thank you so much for being kind and patient with us. Please find below our response to your questions and comments. We have also added our response to the first round of comments just for your perusal. 

Response to Reviewer’s Second Round Comments 

1. Why have keywords been eliminated?

Author Response: Following the PLOS one guideline, we have input keywords directly in the application system. Sorry for the confusion.

 We have revised the keywords using the online “MeSH on Demand” tool. Revised Keywords are: “Breast cancer screening”, “Intention”, “Mammography”, “Breast self-examination”, “Nepal”.

2. In Outcome variable and assessment- Intention to have breast cancer screening part, why was this variable transformed to binary outcome? You will lose the main information with this action. Please do not change this variable and use ordinal logistic regression model unless there is a valid study which previously indicated you can transform this outcome likewise Potential confounders and assessment- Knowledge of breast cancer. Please do not transform the ordinal variables to binary variables. This could reduce the validity of the results of analysis and please check the pre-assumptions of logistic models.

Author Response: We agree that the ordinal outcome variable should have been analyzed with ordinal logistic regression for higher statistical power. However, we dichotomized the outcome variables and employed binary logistic regression because of the following reasons:

1) The questionnaire adopted in this study is a validated questionnaire developed by Godin et al 2001[ref 1 below]. In their paper, Godin et al have also later on dichotomized the intention outcome variable into two categories during their analysis. Similarly, another study by Hart et al 2009 [ref 2 below] has also dichotomized the screening intention outcome variable into two categories and applied binary logistic regression.

2) Moreover, as per our literature review of the studies using TPB, behaviors and intentions are usually defined and measured dichotomously. Every behavior outcomes are measured as performing or not performing the behavior. Similarly, intentions are measured as having or not having an intention. Referring to the previous research in our field, we also aimed to evaluate for two categories of intentions instead of evaluating five categories. 

[ref 1]. Godin G, Gagné C, Maziade J, Moreault L, Beaulieu D, Morel S. Breast cancer: The intention to have a mammography and a clinical breast examination-application of the theory of planned behavior. Psychology and Health. 2001 Jul 1;16(4):423-41.

[ref 2]. Hart SL, Bowen DJ. Sexual orientation and intentions to obtain breast cancer screening. Journal of Women's Health. 2009 Feb 1;18(2):177-85.

We would like to ensure that all the pre-assumptions of logistic models were also checked and confirmed.

3. Please transmit Ethics and Data analysis part before the introduction of your variables and avoid many subtitles for Method parts. This can cause misleading.

Author Response: We have minimized the subtitles for the “Methods” section in the revised manuscript. Please note that referring to the format of recently published cross-sectional studies in PLOS ONE journals, we have kept the “Ethics” and “Data analysis” part just before the “Results” section as it was before.

4. Please state why the median criterion reported for Monthly Income covariate.

Author Response: As mentioned in the manuscript in line number 228, the monthly family income of the participants was diverse. Therefore, the median was used to properly represent the sample characteristics in descriptive statistics. However, while conducting regression analysis, monthly income was treated as a continuous variable. 

5. Please interpret the results using the odds ratio and the coefficients affected significantly the responses.

Author Response: As suggested, we have presented the result of logistic regression using odds ratios(adjusted), confidence interval, and p-value in Table 5. Similarly, in the narrative also, the results have been interpreted using the adjusted odds ratios. The significantly associated factors have been clearly identified using the “*” symbol in Table 5. Please refer to the “Results” section of the manuscript.

6. There is not any path diagram or (DAG) and figure to illustrate the results. Please confirm the results with figures and diagrams.

Author Response: In our cross-sectional design setting, we mainly aimed for statistical association study rather than causal inference or path analysis. We, therefore, would like to be cautious and avoid presenting any causal diagrams such as DAGs without substantial evidence of a true causal effect. Furthermore, in scenarios containing a large number of variables, DAGs become multifaceted, complex, and less informative. However, under the Methods section, we have thoroughly explained all the independent variables, covariates/confounders, and outcome variables. All variables were selected based on prior knowledge, evidence, and theories as cited throughout the manuscript.

---

## [Decision Letter · Decision Letter 2]

2 Nov 2020

PONE-D-20-17423R2

Factors associated with breast cancer screening intention in Kathmandu Valley, Nepal

PLOS ONE

Dear Dr. Ong,

Thank you for submitting your manuscript to PLOS ONE. After careful consideration, we feel that it has merit but does not fully meet PLOS ONE’s publication criteria as it currently stands. Therefore, we invite you to submit a revised version of the manuscript that addresses the points raised during the review process.

Thanks for your revision. However, I and the reviewer believe that you need to reanalyze the data. Therefore, you have the last chance to improve the manuscript. Unfortunately, I will reject it if you submit the amc8uort with further analyzing.

Please understand that it is crucial to address reviewer’s comments or provide a strong justification for not revising.

We look forward to receiving your revised manuscript.

Kind regards,

Amir H. Pakpour, Ph.D.

Academic Editor

PLOS ONE

Reviewers' comments:

Reviewer's Responses to Questions

**Comments to the Author**

1. If the authors have adequately addressed your comments raised in a previous round of review and you feel that this manuscript is now acceptable for publication, you may indicate that here to bypass the “Comments to the Author” section, enter your conflict of interest statement in the “Confidential to Editor” section, and submit your "Accept" recommendation.

Reviewer #1: (No Response)

2. Is the manuscript technically sound, and do the data support the conclusions?

Reviewer #1: No

3. Has the statistical analysis been performed appropriately and rigorously? 

Reviewer #1: No

4. Have the authors made all data underlying the findings in their manuscript fully available?

Reviewer #1: No

5. Is the manuscript presented in an intelligible fashion and written in standard English?

Reviewer #1: Yes

6. Review Comments to the Author

Reviewer #1: Dear Author(s),

Unfortunately, I should inform you that your paper is not acceptable.

I would like you either re analyze your work or present the valid evidence to support your statistical analysis. Unfortunately, your second edition does not have meet the Plos One criteria too and this article did not reach the compulsory merits. I am just able to channel your statistical inference to further research and learning more about using the categorical and dichotomize analysis precisely. It could be useful for future research: https://www.ncbi.nlm.nih.gov/pmc/articles/PMC1977443/pdf/brjcancer00075-0183.pdf

7. PLOS authors have the option to publish the peer review history of their article (what does this mean?). If published, this will include your full peer review and any attached files.

Reviewer #1: No

---

## [Author Response · Author response to Decision Letter 2]

16 Dec 2020

Response to Reviewer and Editor comments

Editor’s latest comment: Thanks for your revision. However, I and the reviewer believe that you need to reanalyze the data. Therefore, you have the last chance to improve the manuscript. Unfortunately, I will reject it if you submit the manuscript without further analyzing.

Please understand that it is crucial to address reviewer’s comments or provide a strong justification for not revising. 

Reviewer’s latest comment: I would like you either re-analyze your work or present the valid evidence to support your statistical analysis. Unfortunately, your second edition does not have to meet the Plos One criteria too and this article did not reach the compulsory merits. I am just able to channel your statistical inference to further research and learning more about using the categorical and dichotomize analysis precisely. It could be useful for future research: https://www.ncbi.nlm.nih.gov/pmc/articles/PMC1977443/pdf/brjcancer00075-0183.pdf

Reviewer’s past comments related to above: In Outcome variable and assessment- Intention to have breast cancer screening part, why was this variable transformed to binary outcome? You will lose the main information with this action. Please do not change this variable and use ordinal logistic regression model unless there is a valid study which previously indicated you can transform this outcome likewise Potential confounders and assessment- Knowledge of breast cancer. Please do not transform the ordinal variables to binary variables. This could reduce the validity of the results of analysis and please check the pre-assumptions of logistic models.

Author Response: Thank you for your detailed review, constructive comments, and suggestion. Concurring with reviewer/editor’s comments, we have reanalyzed the data using ordinal regression. Accordingly, we have revised/updated all affected result tables and narratives of our manuscript. Please kindly refer to “Manuscript with track changes.docx” to easily locate the changes made throughout the manuscript. A general overview of the changes made are described below:

1) In our earlier submitted version of the manuscript, we had transformed the ordinal outcome variable into a binary outcome and employed binary logistic regression. However, in this revised version, we have reanalyzed the data using an ordinal regression model concurring with the reviewer’s suggestion. Please kindly refer to the “Data analysis” section of the revised manuscript (lines ‘205-245’). 

2) In the earlier submitted version of the manuscript, we had dichotomized the ‘knowledge’ and ‘fatalism’ variables as high/low level. However, in this revised version, we have used those variables as ‘continuous variables’ concurring with the reviewer’s suggestion. Please kindly refer to lines ‘153-155,’ and ‘184-185’ of the revised manuscript and also the updated tables- “Table 2” and “Table 5”. 

Extracted changes from the revised manuscript for your convenience:

Line 153-155: “…. The total score for fatalism (range 0-11) was calculated by adding responses of all 11 items which were coded as yes=1 and no=0. Fatalism was treated as a continuous variable and its’ higher value represented higher fatalism. ….”

Line 184-185: “… The total score for knowledge was calculated by adding scores of all 21 items and it was treated as a continuous variable… .”

Line 205-245: “… . Due to the ordinal nature of outcome variables, data were first fitted with the standard ordinal logistic regression (i.e. proportional odds model (POM)) model. However, POM was found inappropriate due to the violation of a proportional odds assumption for some independent variables. The proportional odds assumption was checked using a series of Wald tests, and Brant tests [40, 41]. The other alternatives were PPOM and fully unconstrained generalized ordered logit model (GOLM). PPOM relaxes the proportional odds assumptions for only those variables where it is violated. Whereas, GOLM relaxes the assumption for all variables, even if the assumption was violated by a few of them [42], resulting in too many parameters. Therefore, PPOM is usually considered a more efficient alternative to the GOLM [42]. Nonetheless, both PPOM and GOLM models were compared based on the Akaike Information Criterion (AIC) and Bayes’ Information Criterion (BIC) [43], and the PPOM model was found to have smaller AIC and BIC statistics. We, therefore, used PPOM as the final model to assess the factors associated with the intention to undergo screening. ..”

---

## [Editor Report · Decision Letter 3]

11 Jan 2021

Factors associated with breast cancer screening intention in Kathmandu Valley, Nepal

PONE-D-20-17423R3

Dear Dr. Ong,

We’re pleased to inform you that your manuscript has been judged scientifically suitable for publication and will be formally accepted for publication once it meets all outstanding technical requirements.

Kind regards,

Amir H. Pakpour, Ph.D.

Academic Editor

PLOS ONE
---

## [Editor Report · Acceptance letter]

13 Jan 2021

PONE-D-20-17423R3 

Factors associated with breast cancer screening intention in Kathmandu Valley, Nepal 

Dear Dr. Ong:

I'm pleased to inform you that your manuscript has been deemed suitable for publication in PLOS ONE. Congratulations! Your manuscript is now with our production department. 

Kind regards, 

on behalf of

Dr. Amir H. Pakpour 

Academic Editor

PLOS ONE